# BETANAS: BALANCED TRAINING AND SELECTIVE DROP FOR NEURAL ARCHITECTURE SEARCH

## ABSTRACT

Automatic neural architecture search techniques are becoming increasingly important in machine learning area. Especially, weight sharing methods have shown remarkable potentials on searching good network architectures with few computational resources. However, existing weight sharing methods mainly suffer limitations on searching strategies: these methods either uniformly train all network paths to convergence which introduces conflicts between branches and wastes a large amount of computation on unpromising candidates, or selectively train branches with different frequency which leads to unfair evaluation and comparison among paths. To address these issues, we propose a novel neural architecture search method with balanced training strategy to ensure fair comparisons and a selective drop mechanism to reduce conflicts among candidate paths. The experimental results show that our proposed method can achieve a leading performance of 79.0% on ImageNet under mobile settings, which outperforms other state-of-the-art methods in both accuracy and efficiency.

## 1 INTRODUCTION

The fast developing of artificial intelligence has raised the demand to design powerful neural networks. Automatic neural architecture search methods (Zoph & Le, 2016; Zhong et al., 2018; Pham et al., 2018) have shown great effectiveness in recent years. Among them, methods based on weight sharing (Pham et al., 2018; Liu et al., 2018; Cai et al., 2018; Guo et al., 2019) show great potentials on searching architectures with limited computational resources. These methods are divided into 2 categories: alternatively training ones (Pham et al., 2018; Liu et al., 2018; Cai et al., 2018) and one-shot based ones (Brock et al., 2017; Bender et al., 2018). As shown in Fig 2, both categories construct a super-net to reduce computational complexity. Methods in the first category parameterize the structure of architectures with trainable parameters and alternatively optimize architecture parameters and network parameters. In contrast, one-shot based methods train network parameters to convergence beforehand and then select architectures with fixed parameters. Both categories achieve better performance with significant efficiency improvement than direct search.

Despite of these remarkable achievements, methods in both categories are limited in their searching strategies. In alternatively training methods, network parameters in different branches are applied with different training frequency or updating strength according to searching strategies, which makes different sub-network convergent to different extent. Therefore the performance of sub-networks extracted from super-net can not reflect the actual ability of that trained independently without weight sharing. Moreover, some paths might achieve better performance at early steps while perform not well when actually trained to convergence. In alternatively training methods, these operators will get more training opportunity than other candidates at early steps due to their well performance. Sufficient training in turn makes them perform better and further obtain more training opportunities, forming the Matthew Effect. In contrast, other candidates will be always trained insufficiently and can never show their real ability.

Differently, One-shot methods train paths with roughly equal frequency or strength to avoid the Matthew Effect between parameters training and architectures selection. However, training all paths to convergence costs multiple time. Besides, the operators are shared by plenty of sub-networks, making the backward gradients from different training steps heavily conflict. To address this issue,

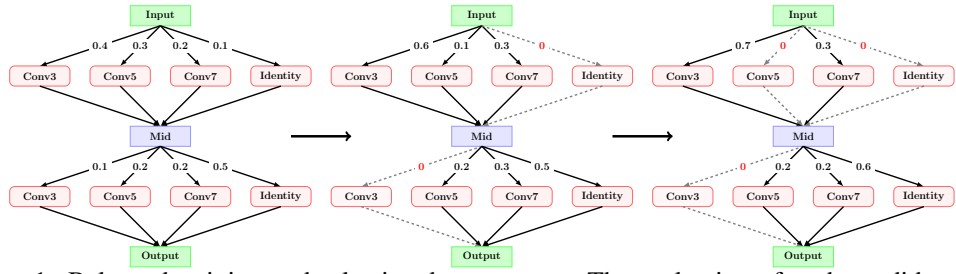

Figure 1: Balanced training and selective drop strategy. The evaluation of each candidate only influence architecture search strategy, and paths with low performance will be gradually dropped to reduce conflicts among paths. In addition, paths still remaining are all trained with comparable frequency to insure a fair comparison among candidate operators.

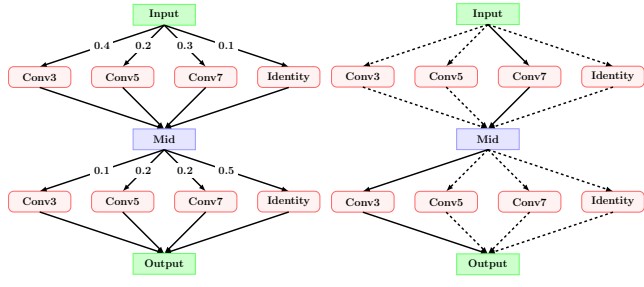

(a) alternatively training methods    (b) one-shot methods

Figure 2: Weight sharing methods. In alternatively training methods, candidate paths with better performance get more training opportunities or higher fusion weight. In one-shot methods, sub-networks are uniformly randomly trained to convergence before architecture selection.

we follows the balanced training strategy to avoid the Matthew Effect, and propose a drop paths approach to reduce mutual interference among paths, as shown in Fig 1.

Experiments are conducted on ImageNet classification task. The searching process costs less computational resources than competing methods and our searched architecture achieves an outstanding accuracy of 79.0%, which outperforms state-of-the-art methods under mobile settings. The proposed method is compared with other competing algorithms with visualized analysis, which demonstrates its the effectiveness. Moreover, we also conduct experiments to analysis the mutual interference in weight sharing and demonstrate the rationality of the gradually drop paths strategy.

## 2    RELATED WORK

Automatic neural architecture search techniques has attracted much attention in recent years. NAS-Net (Zoph & Le, 2016; Zoph et al., 2018) proposes a framework to search for architectures with reinforcement learning, and evaluates each of the searched architectures by training it from scratch. BlockQNN (Zhong et al., 2018; Guo et al., 2018; Zhong et al., 2018) expands the search space to the entire DAG and selects nets with Q-learning. Network pruning methods (Li et al., 2019; Noy et al., 2019) prune redundant architectures to reduces search spaces. Considering the searching policy, most of these methods depend on reinforcement learning, evolutionary algorithms and gradient based algorithms (Bello et al., 2017; Liu et al., 2018; Cai et al., 2018).

The most related works to our method are the ones based on weight sharing proposed by (Pham et al., 2018), from which two streams are derived: Alternatively training methods (Cai et al., 2018; Liu et al., 2018) and one-shot methods (Brock et al., 2017; Bender et al., 2018; Guo et al., 2019). Methods in the first stream alternatively train architecture parameters and network parameters. During search process, operators in the super-net are selectively trained and evaluated with a certain policy and the policy is updated dynamically according to the evaluations. Among them, ENAS (Pham et al., 2018) introduces RL to select paths. DARTS (Liu et al., 2018) improves the accuracy and efficiency of paths selection policy by considering the importance of each path as trainable parameters. ProxyLessNAS (Cai et al., 2018) proposes to directly search on target datasets with single paths and makes the latency term differentiable. Single-Path NAS (Stamoulis et al., 2019) directly shares

weights via super-kernel. By contrast, one-shot based methods Guo et al. (2019); Brock et al. (2017); Bender et al. (2018) firstly train each path in the super-net with equal frequency to convergence, then all the architectures are selected from the super-net and evaluated with fixed parameters. Darts+ Liang et al. (2019) improves Darts with early stop. Progressive-NAS Chen et al. (2019) gradually increases blocks while searching. HM-NAS Yan et al. (2019) uses mask to select paths. (Sciuto et al., 2019) Our work benefits from the advantages of both categories: On one hand, the importance factors are evaluated with a gradient based approach but it has no influence on training shared parameters. On the other hand, the shared parameters are updated uniformly as those in one-shot.

## 3 APPROACH

ProxyLessNAS(Cai et al., 2018) and Single Path One-shot(Guo et al., 2019) proposed to train the super-net with only one path on in each step to make the performance trained with weight sharing more close to that trained alone. Both of them enhance the performance of weight sharing to a higher level. ProxyLessNAS updates architecture parameters and network parameters alternatively. Paths are selectively trained according to their performance, and paths with higher performance get more training opportunities. Single Path One-shot first proposed to balanced train all paths until convergence and then use evolution algorithms to select network structures. The equivalent functions of the choice blocks in two methods are described as $m^{PL}$ and $m^{OS}$ in Eq 1:

$$m^{PL}(x) = \begin{cases} o_1(x) & \text{with probability } p_1, \\ \dots, \\ o_N(x) & \text{with probability } p_2. \end{cases}, m^{OS}(x) = \begin{cases} o_1(x) & \text{with probability } 1/N, \\ ..., \\ o_N(x) & \text{with probability } 1/N. \end{cases} \quad (1)$$

Our method follows the alternatively training ones, in which architecture parameters and network parameters are optimized alternatively in each step. To give a better solution to the problems discussed above, we train each candidate path with equal frequency to avoid the "Matthew effect" and gradually dropped least promising paths during searching process to reduce conflicts among candidate paths.

### 3.1 PIPELINE

The pipeline of our method is shown in Algorithm 1. First of all, a super-net is constructed with $L$ choice blocks $O_1, O_2, \dots, O_L$, as shown in Fig 1. Each choice block $O_l$ is composed of $M$ candidate paths and corresponding operators $o_{l,1}, o_{l,2}, \dots, o_{l,M}$. The importance factor of $o_{l,m}$ is denoted as $\alpha_{l,m}$ and $\alpha_{l,m}$ are converted to probability factor $p_{l,m}$ using softmax normalization.

Secondly, the parameters of $o_{l,m}$ and their importance factors $\alpha_{l,m}$ are trained alternatively in **Phase 1** and **Phase 2**. When training $\alpha_{l,m}$, latency term is introduced to balance accuracy and complexity. Paths with $\alpha_{l,m}$ lower than $th_\alpha$ will be dropped and no more trained.

---

**Algorithm 1** Searching Process

**Initialization:** Denote $O_l$ as the choice block for layer $l$ with $M$ candidate operators $\{o_{l,1}, o_{l,2}, \dots, o_{l,M}\}$. $\alpha_{l,1}, \alpha_{l,2}, \dots, \alpha_{l,M}$ are the corresponding importance factors of candidate operators and initialized with identical value. $S_{max}$ denotes the max number of optimization steps.

1: **while** $t < S_{max}$ **do**
2:     **Phase1:** Randomly select $o_{l,m_l} \in O_l$ for block $O_l$ with uniform probability, then fix all $\alpha_{l,m}$ and train the super-net constructed with the selected $o_{1,m_1}, o_{2,m_2}, \dots, o_{L,m_L}$ for some steps.
3:     **Phase2:** Fix all the parameters in $o_{l,m}$ and measure their flops/latency. Then evaluate each operator $o_{l,m}$ with both cross-entropy loss and flops/latency loss. Update $\alpha_{l,m}$ according to the losses feedback.
4:     **for** $o_{l,m} \in O_l$ **do**
5:         **if** $\alpha_{l,m} < th_\alpha$ **then** $O_l = O_l \setminus \{o_{l,m}\}$
        $t = t + 1$
6: **for** $o_{l,m} \in O_l$ **do** $m_l = argmax_m(\alpha_{l,m})$
7: **return** $o_{1,m_1}, o_{2,m_2}, \dots, o_{L,m_L}$

---

Finally, after alternatively training $o_{l,m}$ and $\alpha_{l,m}$ for given steps, paths with the highest importance factor in each choice block are selected to compose a neural architecture as the searching result.

## 3.2 BALANCED TRAINING

Alternatively training methods focus computational resources on most promising candidates to reduce the interference from redundant branches. However, some operators that perform well at early phases might not perform as well when they are trained to convergence. These operators might get much more training opportunities than others due to their better performance at the beginning steps. Higher training frequency in turn maintains their dominant position in the following searching process regardless their actual ability, forming the Matthew Effect. In contrast, the operators with high performance when convergent might never get opportunities to trained sufficiently. Therefore, the accuracy of alternatively training methods might degrade due to inaccurate evaluations and comparison among candidate operators.

Our method follows the alternatively optimizing strategy. Differently, we only adopt gradient to architectures optimization while randomly sample paths with uniformly probability when training network parameters to avoid the Matthew Effect. More specifically, when updating network parameters of $o_{l,m}$ in Phase 1 and architecture parameters in Phase 2, the equivalent output of choice block $O_l$ is given as $O_l^{path}$ in Eq 2 and $O_l^{arch}$ in Eq 3:

$$O_l^{path}(x) = o_{l,m}(x) \begin{cases} \text{with probability } \frac{1}{M'} & \text{, if } \alpha_{l,m} > th_\alpha \\ \text{with probability } 0 & \text{, else.} \end{cases} \tag{2}$$

$$O_l^{arch}(x) = o_{l,m}(x) \begin{cases} \text{with probability } p_{l,m} & \text{, if } \alpha_{l,m} > th_\alpha \\ \text{with probability } 0 & \text{, else.} \end{cases} \tag{3}$$

Where $M'$ is the number of remaining operators in $O_l$ currently, and $p_{l,m}$ is the softmax form of $\alpha_{l,m}$. The $\alpha_{l,m}$ of dropped paths are not taken into account when calculating $p_{l,m}$. The parameters in both phases are optimized with Stochastic Gradient Descent (SGD). In Phase 1, the outputs in Eq 2 only depends on network parameters, thus gradients can be calculated with the Chain Rule. In Phase 2, the outputs not only depend on the fixed network parameters but also architecture parameters $\alpha_{l,m}$. Note that $O_l^{arch}(x)$ is not differentiable with respect to $\alpha_{l,m}$, thus we introduce the manually defined derivatives proposed by Cai et al. (2018) to deal with this issue: Eq 3 can be expressed as $O_l^{arch}(x) = \sum g_{l,m} \cdot o_{l,m}(x)$, where $g_{l,0}, g_{l,0}, \dots, g_{l,M'}$ is a one-hot vector with only one element equals to 1 while others equal to 0. Assuming $\partial g_{l,j}/\partial p_{l,j} \approx 1$ according to Cai et al. (2018), the derivatives of $O_l^{arch}(x)$ w.r.t. $\alpha_{l,m}$ are defined as :

$$\begin{aligned} \frac{\partial O_l^{arch}(x)}{\partial \alpha_{l,m}} &= \sum_{j=1}^{M'} \frac{\partial O_l^{arch}(x)}{\partial g_{l,j}} \frac{\partial g_{l,j}}{\partial p_{l,j}} \frac{\partial p_{l,j}}{\partial \alpha_{l,m}} \approx \sum_{j=1}^{M'} \frac{\partial O_l^{arch}(x)}{\partial g_{l,j}} \frac{\partial p_{l,j}}{\partial \alpha_{l,m}} \\ &= \sum_{j=1}^{M'} \frac{\partial O_l^{arch}(x)}{\partial g_{l,j}} p_j(\delta_{mj} - p_m) \end{aligned} \tag{4}$$

From now on, $O_l^{path}(x)$ and $O_l^{arch}(x)$ are differentiable w.r.t. network parameters and architecture parameters respectively. Both parameters can be optimized alternatively in Phase 1 and Phase 2.

## 3.3 SELECTIVELY DROP PATHS

One-shot based methods, such as Single Path One-shot Guo et al. (2019) also uniformly train paths. These methods train network parameters of each path uniformly to convergence, after that a searching policy is applied to explore a best structure with fixed network parameters. However, the optimizations of candidate operators in a same choice block actually conflict. Considering $N$ candidate operators in a same choice block and their equivalent functions $f_1, f_2, \dots, f_N$, given $F_{in}$ and $F_{out}$ as the equivalent functions of the sub-supernet before and after the current choice block, $x_i$ and $y_i$ as input

data and labels from the training dataset, and $\mathcal{L}$ as the loss metric, the optimization of network parameters can be described as:

$$\min_{w_n} \mathcal{L}(y_i, F_{out}(f_n(F_{in}(x_i), w_n))), n = 1, 2, \ldots, N \tag{5}$$

When the super-net is trained to convergence, $F_{in}$ and $F_{out}$ is comparatively stable. In this situation, $f_1(w_1), f_2(w_2), \ldots, f_N(w_N)$ are actually trained to fit a same function. However when operators are trained independently without weight sharing, different operators are unlikely to output same feature maps. Take the super-net in Fig 2(b) as an example, the four operators in choice block 1 are likely to be optimized to fit each other. Intuitively, Conv3 and Identity are unlikely to fit a same function when trained without weight sharing. On the other hand, the operators in the second choice block are trained to be compatible with various input features from different operators in the first choice block. In contrast, each operator process data from only one input when networks are trained independently. Both problems widen the gap between network trained with and without weight sharing.

Fewer candidate paths help reduce the conflicts among operators, which will explained in the experiments section. Therefore, a drop paths strategy is applied in our method to reduce mutual interference among candidate operators during searching process. The paths with performance lower than a threshold will be permanently dropped to reduce its influence on remaining candidates.

When updating $\alpha_{l,m}$ in phase 2, we follow the strategy in ProxyLessNAS (Cai et al., 2018) to sample path in each choice block with probability $p_{l,m}$, and optimize $\alpha_{l,m}$ by minimizing the expectation joint loss $\mathcal{L}$:

$$\mathcal{L} = \mathcal{L}_{CE} + \beta\mathcal{L}_{LA} = \mathcal{L}_{CE} + \beta \sum_{l=1}^{\mathcal{L}} \sum_{m=1}^{M} p_{l,m}\mathcal{L}_{LA_{l,m}} \tag{6}$$

where $\mathcal{L}_{CE}$ and $\mathcal{L}_{LA}$ are the expectation cross-entropy loss and latency loss, $\mathcal{L}_{LA_{l,m}}$ is the flops or latency of operator $o_{l,m}$ and $\beta$ is a hyper-parameter to balance two loss terms. We regard the importance factor $\alpha_{l,m}$ and its softmax form $p_{l,m}$ as sampling probability and use Eq 6 to optimize $\alpha_{l,m}$. The derivatives of $\mathcal{L}_{CE}$ and $\mathcal{L}_{LA}$ w.r.t. $\alpha_{l,m}$ can be get from Eq 4 and 6 respectively. Note that $\alpha_{l,m}$ is only applied to evaluate the importance of paths and have no influence on the balanced training strategy in Phase 1. After each step of evaluation, paths with low $\alpha_{l,m}$ are dropped and will not be trained anymore:

$$O_l = O_l \setminus \{o_{l,m_l}, \text{if } \alpha_{l,m_l} < th_\alpha, \forall m_l\} \tag{7}$$

The limitations of alternatively training methods and one-shot based ones are relieved by the proposed balanced training and drop paths strategies respectively. The two phases are trained alternatively until meeting the stop condition, then paths with highest $\alpha_{l,m}$ from each block are selected to compose an architecture as the searching result.

## 4 EXPERIMENTS

### 4.1 EXPERIMENTAL SETTINGS

**Datasets** The target architecture is directly searched on ImageNet (Deng et al., 2009) for classification task. 50000 images are extracted from training set to train architecture parameters $\alpha_{l,m}$, and the rest of training dataset is used to train network weights.

**Search Space** We follow the search space in MNASNet (Tan et al., 2018), where each choice block includes an identity operator and 6 MobileNetV2 blocks which have kernel size 3,5,7 and expand ratio 3, 6 respectively. SE-Layer (Hu et al., 2018) is not applied to the search space.

**Training Detail** We search on V100 GPUs for 160 GPU hours. The shared parameters are trained with 1024 batch size and 0.1 learning rate. $\alpha_{l,m}$ is trained with Adam optimizer and 1e-3 initial learning rate. Finally, the searched architecture is trained from scratch according to the setting of MobileNetV2 (Sandler et al., 2018). The searched networks are trained from scratch on training dataset with hyper-parameters as follows: batch size 2048, learning rate 1.4, weight decay 2e-5, cosine learing rate for 350 epochs, dropout 0.2, label smoothing rate 0.1.

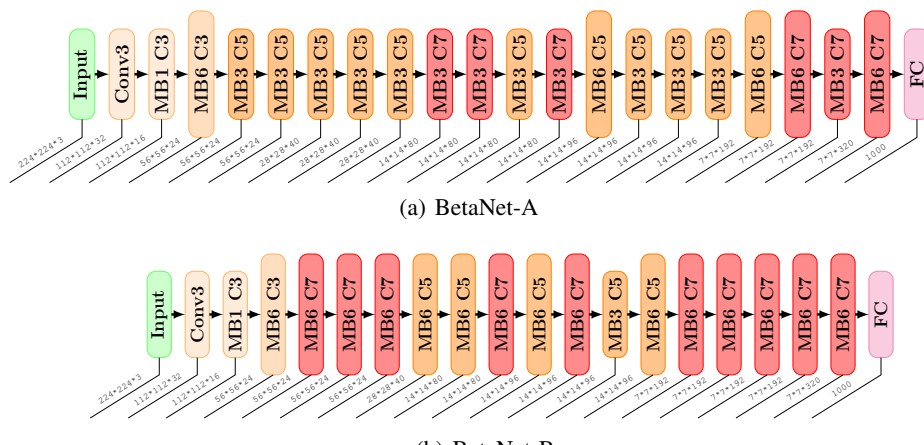

(a) BetaNet-A

(b) BetaNet-B

Figure 3: The searched architectures with different complexity limitations: BetaNet-A is searched with flops limitation. BetaNet-B is searched with latency limitaion.

## 4.2 EXPERIMENTAL RESULTS

Table 1: BetaNet-A compared with the state-of-the-art methods under comparable flops.

| Method | Top1 (%) | Top5 (%) | Flops (M) | params (M) | cost (GPU*h) |
|---|---|---|---|---|---|
| MobileNet V2 (Sandler et al., 2018) | 72.0 | 91.0 | 300 | 3.4 | - |
| DARTS (Liu et al., 2018) | 73.1 | 91.0 | 595 | 4.9 | 96 |
| FBNet-C (Wu et al., 2018) | 74.9 | - | 375 | - | - |
| ProxyLessNAS (Cai et al., 2018) | 74.6 | 92.2 | 320 | - | 200 |
| MNASNet+SE (Tan et al., 2018) | 75.4 | 92.5 | 317 | 4.2 | 40,000 |
| MobileV3+SE+swish (Howard et al., 2019) | 75.2 | - | 219 | 5.4 | - |
| **BetaNet-A** | 75.1 | 92.3 | 315 | 3.6 | 160 |
| **BetaNet-A + SE** | **75.9** | 92.8 | 333 | 4.1 | 160 |
| MobileNet V2 (Sandler et al., 2018) × 1.4 | 74.7 | 92.5 | 585 | 6.9 | - |
| EffNetB0+SE+swish+autoaug Tan & Le (2019) | 76.3 | 93.2 | 390 | 5.3 | - |
| EffNetB1+SE+swish+autoaug Tan & Le (2019) | 78.8 | 94.4 | 700 | 7.8 | - |
| **BetaNet-A × 1.4** | **77.1** | 93.5 | 596 | 6.2 | 160 |
| MNASNet× 1.4 + SE (Tan et al., 2018) | 77.2 | 93.2 | 600 | - | 40000 |
| **BetaNet-A× 1.4 + SE** | **77.7** | 93.7 | 631 | 7.2 | 160 |
| **BetaNet-A× 1.4 + SE + auto-aug + swish** | **79.0** | 94.2 | 631 | 7.2 | 160 |

Table 2: BetaNet-B compared with the state-of-the-art methods under comparable GPU latency.

| Method | Top1 (%) | Top5 (%) | GPU Latency (ms) | search cost (GPU hours) |
|---|---|---|---|---|
| MobileNet V2 (Sandler et al., 2018) | 72.0 | 91.0 | 6.1 | - |
| ShuffleNetV2 (1.5) (Ma et al., 2018) | 72.6 | - | 7.3 | - |
| ProxyLessNAS-gpu (Cai et al., 2018) | 75.1 | 92.5 | 5.1 | 200 |
| MNASNet (Tan et al., 2018) | 74.0 | 91.8 | 6.1 | 40,000 |
| **BetaNet-B** | **75.8** | 92.8 | 6.2 | 160 |

**Searched Archtectures** As shown in Fig 3, BetaNet-A and BetaNet-B are searched with flops and GPU latency limitation respectively. BetaNet-A tends to select operators with lower flops at front layers where feature maps are large and operators with higher flops elsewhere to enhance the ability. BetaNet-B tends to select large kernels and fewer layers, since GPU performs better with parallelism.

**Performance on ImageNet** Experiments results compared with state-of-the-art methods under comparable flops and gpu latency are shown in Table 1 and 2 respectivly and our architectures

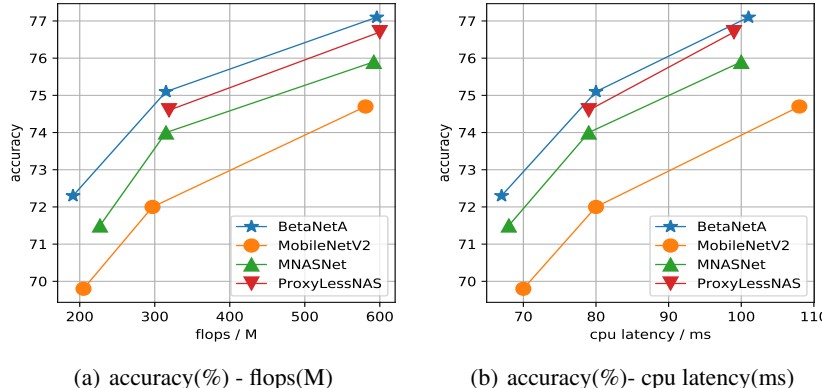

(a) accuracy(%) - flops(M)          (b) accuracy(%)- cpu latency(ms)

Figure 4: BetaNet-A is compared with MobileNetV2 (Sandler et al., 2018) ProxyLessNAS Cai et al. (2018) and MNASNet (Tan et al., 2018) with various depth multiplier (0.75, 1.0, 1.4).

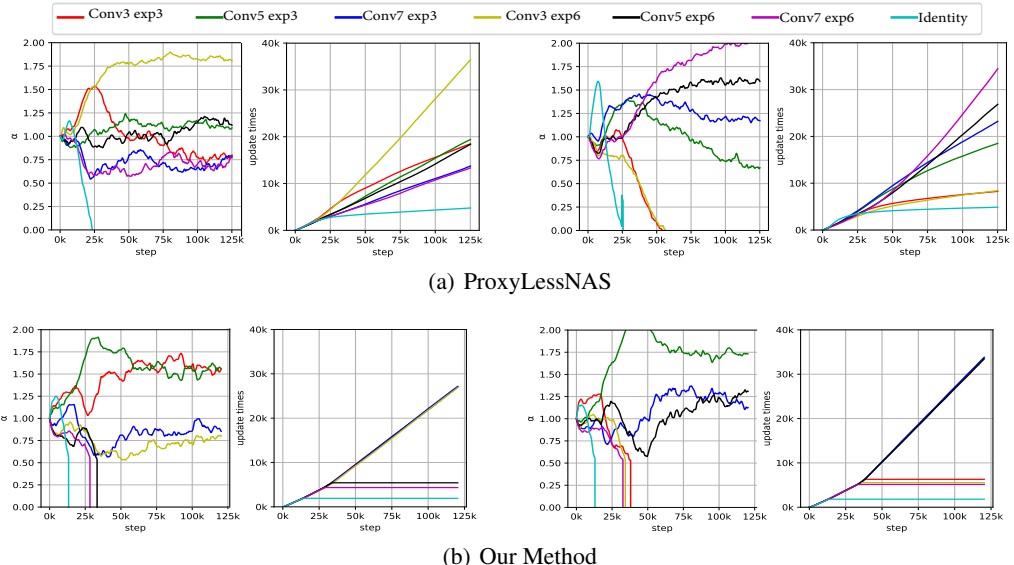

Figure 5: The variation of $\alpha$ and training frequency of 2 choice blocks from each method are shown in (a) and (b). From left to right in each line: $\alpha$ in block 1, paths training frequency in block 1, $\alpha$ in block 2, paths training frequency in block 2.

achieve the best performance. SENet (Hu et al., 2018) with ratio $0.0625$ is applied in table 1 as BetaNet-A + SE. BetaNet-A performs better than MobileNetV2 with comparable flops by 3.1%. BetaNet-B performs better with comparable latency by 3.8%.Auto-augment (Cubuk et al., 2018) and SWISH activation (Ramachandran et al., 2017) are also applied to the searched BetaNet-A and the performance is further enhanced to 79.0%. As shown in Fig 4, BetaNet-A outperforms MobileNetV2, ProxyLessNAS (Cai et al., 2018) and MNASNet (Tan et al., 2018) with various depth multiplier.

## 4.3 ABLATION STUDIES

### 4.3.1 COMPARED WITH GRADIENT BASED METHODS

Experiments are conducted in this sub-section to analyze the contribution of the proposed searching approach: balanced training and selective drop. The searching process of BetaNet-A is compared with that of ProxyLessNAS (Cai et al., 2018). Fig 5(a) and (b) show the variation of architecture parameters $\alpha$ and updating frequency of 2 choice blocks for each method respectively. In the result of ProxyLessNAS shown in Fig 5(a) (top left), the conv3_exp6 operator performs better at early phases and get much more training opportunity than other operators and thus trained more sufficiently, which might lead to degraded performance due to the above mentioned "Matthew Effect". Differently, our strategy ensures that all remaining paths are trained with roughly equal frequency. In addition, our

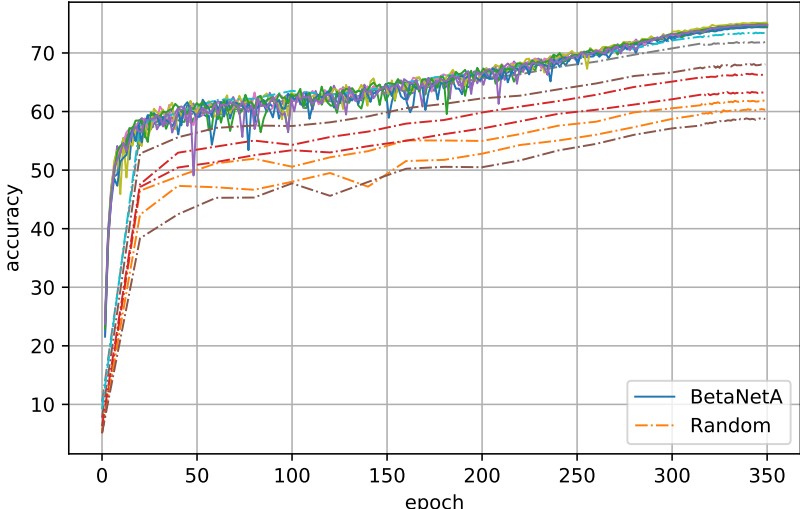

Figure 6: Comparison between architectures searched with different random seeds and randomly sampled ones. Architectures searched with our method and with random policy are shown with solid lines and dotted lines respectively.

methods train remaining with much more frequency than ProxyLessNAS, while ProxyLessNAS spend much steps on redundant paths with little promising.

### 4.3.2 EXPERIMENTS WITH DIFFERENT RANDOM SEEDS

(Sciuto et al., 2019) suggests to evaluate weight sharing methods with different random seeds. To examine the robustness of the proposed method, experiments are conducted with our approach with 8 random seeds. 8 architectures are searched independently. We also randomly sample 8 architectures from the same search space and super-net. Constraint is applied to insure both searched and randomly sampled networks have similar flops with BetaNetA. All the 16 networks are trained with the same setting. As shown in Fig 6, all the searched architectures have similar performance on ImageNet. Our method outperforms the random sampled ones and other competing methods with robust performance.

### 4.3.3 ANALYSIS OF DROP PATH

Besides the policy of selecting architecture parameters, the optimization of network parameters of our method is different from that of one-shot methods mainly in the drop paths strategy. To give a better understanding of its effectiveness, we conduct experiments on 4 network architectures with 3 different training policies to explore the influence of the number of candidate branches on weight sharing. As shown in Figure7, networks in 3 groups are trained with none weight sharing(NS), 2-branch weight sharing branches(B2) and 4-branch weight sharing branches(B4), respectively. Networks in NS, B2, B4 are trained on cifar10 for 30, 60, 120 epochs respectively to insure network parameters are trained with comparable steps in each group.

The accuracy of the 4 networks trained with NS, B2 and B4 are shown in Fig7 respectively. The experiments indicate 2 phenomenons: 1. The accuracy trained via B2 with 60 epochs is much higher than B4 with 120 epochs, which indicates that less branches in weight sharing helps network parameters converge better. 2. The relative rank of the 4 networks trained with B2 is more similar to those with NS than those with B4, which indicates that less branch can give a better instruction in network selection and demonstrate the rationality of our drop paths strategy.

### 4.3.4 COMPARISON WITH OTHER WEIGHT SHARING METHODS ON SEARCH STRATEGIES

Our searching and optimizing strategy is compared to those of other weight sharing approaches, as shown in Tab 3. Alternatively training methods tend to adopt similar strategies to decide which path to train in optimizing phases and which path to evaluate in search phases. However searching strategies are designed to select paths and inevitably treat candidates discriminatively. Therefore,

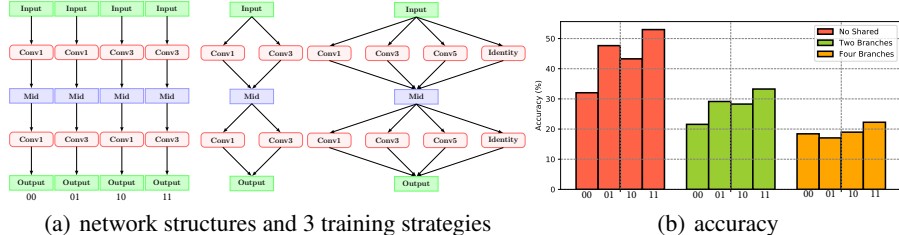

(a) network structures and 3 training strategies      (b) accuracy

Figure 7: 4 Networks are trained in 3 groups with different training policies. 00, 01, 10, 11 represent architectures conv1-conv1, conv1-conv3, conv3-conv1, conv3-conv3 respectively.

Table 3: Comparison with the other weight sharing methods on their strategies.

| Method | supernet optimization | search policy | sample sub-net | mutual interference |
|---|---|---|---|---|
| ENAS Pham et al. (2018) | imbalanced | RL | $\checkmark$ | all candidates |
| Darts Liu et al. (2018) | imbalanced | gradient | $\times$ | all candidates |
| SinglePath Guo et al. (2019) | balanced | EA | $\checkmark$ | all candidates |
| ProxyLess Cai et al. (2018) | imbalanced | gradient/RL | $\checkmark$ | all candidates |
| Progressive Chen et al. (2019) | imbalanced | gradient | $\times$ | fewer candidates |
| Darts+ Liang et al. (2019) | imbalanced | gradient | $\times$ | all candidates |
| HM-NAS Yan et al. (2019) | imbalanced | gradient+mask | $\times$ | all candidates |
| Ours | balanced | gradient | $\checkmark$ | fewer candidates |

searching strategies are usually not suitable for optimizing phases. Our method applies different policies in optimizing phases and searching phases to deal with this problem.

Some of the methods sample a single sub-net in each step while training super-net instead of summing up features from all possible paths to bridge the gap between training super-net and evaluating sub-nets. Our methods follow this strategy to improve performance and reduce GPU memory consumption as well. To deal with mutual interference among candidates in a same choice block. Progressive-Darts Chen et al. (2019) and our method make effort to drop redundant paths to reduce the interference from them.

### 4.4 DISCUSSION ON TWO STREAMS OF WEIGHT SHARING METHODS

There is always a compromise between accuracy and efficiency. Weight sharing shows remarkable improvements in reducing searching cost though introduce inevitable bias on accuracy. Alternatively training approaches are actually greedy methods, since they are talented at searching for next optimal solution. Meanwhile, architectures with less competition at early phases are abandoned. In contrast, one-shot methods attempt offer equal opportunities to all candidates which leads to a more global solution. However operators via weight sharing should deal with outputs from multiple former paths, which is challenging for operators with less flops. Therefore these operators suffer more from the mutual interference than those with larger flops.

Our approach tries to balance the advantages and disadvantages of both streams. On one hand, we tries to insure the accuracy of most promising operators which is similar to the strategies in alternatively training ones. By contrast, only the operators with performance much lower than the average of others will be dropped. On the other hand, we train paths balanced follows the strategies one-shot based ones. Unlikely, paths are gradually dropped to a lower amount in our method to reduce conflicts.

## 5 CONCLUSION

This work proposes a novel neural architecture search method via balanced training and selective drop strategies. The proposed methods benefits from both streams of weight sharing approaches and relieve their limitations in optimizing the parameters in super-net. Moreover, our method achieves a

new state-of-the-art result of 79.0% on ImageNet under mobile settings with even less searching cost, which demonstrates its effectiveness.

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
