# OpenReview forum: "BETANAS: Balanced Training and selective drop for Neural Architecture Search"
_ICLR.cc/2020/Conference — Reject_

### Official Review · AnonReviewer1 · 2019-10-24
**Official Blind Review #1**

**Rating:** 3

**Review:**

This paper introduces a better searching strategy in the context of automatic neural architecture search (NAS). Especially, they focus on improving the search strategy for previously proposed computationally effective weight sharing methods for NAS. Current search strategies for the weight sharing NAS methods either focus on uniformly training all the network paths or selectively train different network paths with different frequency, where both have their own issues like wasting resources for unpromising candidates and unfair comparison among network paths. To this end, this paper proposes a balanced training strategy with “selective drop mechanism”. Further, they validate their approach by showing leading performance on ImageNet under mobile settings.

Overall, I appreciate the effort on exploring new strategies for better search algorithm for NAS in the context of weight sharing methods. However, the analysis of the current approach is limited and some relevant papers are missing. More details below.

Arguments:
1) This paper’s main focus is on developing a better search strategy for NAS based weight sharing methods. The validation of their approach is supported via showing improvement in the accuracy on ImageNet task. However, my main argument is that you have to validate the proposed approach with better analysis on the search space along with improvement on the tasks. For example, “Evaluating the Search Phase of Neural Architecture Search” paper which studies the effectiveness of current search strategies is not referenced in this paper. I would suggest to do analysis of you search strategy by following the analysis experiments conducted in the above paper.

2) The related work doesn’t discuss the latest papers which suggest why and in what ways the current search strategies are based for weight-sharing based NAS approaches. As this paper is trying to address this problem, I would naturally assume that this is well discussed, but it’s missing!


**Experience Assessment:**

I have published one or two papers in this area.

**Review Assessment: Checking Correctness Of Derivations And Theory:**

I assessed the sensibility of the derivations and theory.

**Review Assessment: Checking Correctness Of Experiments:**

I assessed the sensibility of the experiments.

**Review Assessment: Thoroughness In Paper Reading:**

I read the paper at least twice and used my best judgement in assessing the paper.

---

> ### Author Response · Authors · 2019-11-11
> **Response to Reviewer #1**
>
> Thanks for your instructive comments and suggestion.
> 1)
> We have updated our paper with evaluation experiments according to "Evaluating the Search Phase of Neural Architecture Search". More experiments with different random seeds are conducted to compare our methods with randomly sampled architectures from the same search space. Our method outperforms the randomly sampled ones and other competing methods, as shown in Section 4.3.2 and Fig.6.
>
> 2) The suggestion is meaningful and instructive. We have made a discussion on recently weight sharing methods in Tab(3) and Section 4.3.4, to compare their training approaches, searching strategies, efficiency, etc. These papers and "Evaluating ..." are introduced to related works and analyzed in Experiments and Discussion sections.
>
> Table 3. Comparison with the other weight sharing methods on their strategies.
> +----------------+--------------------------------+--------------------+-----------------------+--------------------------------+
> | Method      | supernet optimization | search policy | sample sub-net | mutual interference     |
> +----------------+--------------------------------+--------------------+-----------------------+--------------------------------+
> | ENAS          | imbalanced                    | RL                     | √                           | from all candidates      |
> +----------------+--------------------------------+--------------------+-----------------------+--------------------------------+
> | Darts          | imbalanced                    | gradient          | x                           | from all candidates      |
> +----------------+--------------------------------+--------------------+-----------------------+--------------------------------+
> | SinglePath | balanced                        | EA                     | √                           | from all candidates      |
> +----------------+--------------------------------+--------------------+-----------------------+--------------------------------+
> | ProxyLess  | imbalanced                    | gradient/RL    | √                           | from all candidates      |
> +----------------+--------------------------------+--------------------+-----------------------+--------------------------------+
> | Progressive| imbalanced                   | gradient         | x                            | from fewer candidates|
> +----------------+--------------------------------+--------------------+-----------------------+--------------------------------+
> | Darts+        | imbalanced                    | gradient          | x                           | from all candidates      |
> +----------------+--------------------------------+--------------------+-----------------------+--------------------------------+
> | HM-NAS     | imbalanced                   | gradient+mask| x                          | from all candidates      |
> +----------------+--------------------------------+--------------------+-----------------------+--------------------------------+
> | Ours           | balanced                        | gradient           | √                          | from fewer candidates |
> +----------------+--------------------------------+--------------------+-----------------------+--------------------------------+

---

### Official Review · AnonReviewer2 · 2019-10-27
**Official Blind Review #2**

**Rating:** 3

**Review:**

In this paper, the authors proposed a new training strategy in achieving better balance between training efficiency and evaluation accuracy with weight sharing-based NAS algorithms. It is consisted of two phrases: in phrase 1, all path are uniformly trained to avoid bias, in phrase 2, less competitive options are pruned to save cost. The proposed method achieved the SOTA on IN mobile setting.


Overall I found the idea proposed in the paper intuitive and convincing. Especially I appreciate the ablation study that identified one of the option is encouraged too much in the early stage will can lead to worse final performance. From the methodology perspective, I think this is a solid incremental contribution. However, the highlight of this paper, in my opinion, is the SOTA results on IN. My main concern is that the authors did not indicate that the code/model will be open sourced, which will help verification as well as reproducibility.

**Experience Assessment:**

I have read many papers in this area.

**Review Assessment: Checking Correctness Of Derivations And Theory:**

I did not assess the derivations or theory.

**Review Assessment: Checking Correctness Of Experiments:**

I assessed the sensibility of the experiments.

**Review Assessment: Thoroughness In Paper Reading:**

I made a quick assessment of this paper.

---

> ### Author Response · Authors · 2019-11-11
> **Response to Reviewer #2**
>
> Thanks for your meaningful comments and suggestion. The model architectures, weights and training hyper-parameters of the reported BetaNet-A and BetaNet-B will be open sourced in Google Drive in 2 ~3 days. We believe that our method is easy to follow and re-implement.

---

> > ### Author Response · Authors · 2019-11-15
> > **Additional Response to Reviewer #2**
> >
> > We're sorry to say that the code and models could not be open sourced until the approaval from our organization. We have filed the application for releasing code and models with the legal affairs department. This procedure might cost several weeks. Our training curves and experiments settings for all reported models have been uploaded on Github(  https://github.com/BetanasICLR2020/BetaNAS   ) in details. Other materials will be updated once they are permitted. We believe that our method is easy to follow and re-implement. If anyone has some questions on the method and experiments, we will try our best to help other researchers to re-implement the method.

---

### Official Review · AnonReviewer3 · 2019-11-04
**Official Blind Review #3**

**Rating:** 6

**Review:**

This paper relates to automatic neural architecture search techniques. Current methods have certain drawbacks: Some train all network paths to convergence, which wastes computational efforts in unpromising paths, whereas some don't train all the branches uniformly, which can lead to unfair comparisons.

The authors propose a model to balance the two issues mentioned above. Their aim is to produced balanced training while trying to reduce conflicts between the different potential network paths.  So their algorithm Has two phases, one where it randomly builds a block in the network and another one where it discards vertices form the layer that are below a certain threshold.

The experimental results seem sound to me, and I think this is a reasonable approach.

Here some general comments that would help with clarity:

1) I think the authors should explain more clearly what "Matthew Effect" in the introduction

2) It's not very clear to me how th_\alpha is computed. Could this please me made more specific. Section 4.4 says that "only the operators with performance much lower than the average of others will be dropped." Is this approach conservative? Did they try different thresholds?

There are several minor typos that the authors might want to correct.

1) There is a couple of spaces missing like between "2018)have shown" in page 1
    - The word "probability" and p_1, p_2 in (1)
    - Eq6 on page 5

2) It is customary to use commas before and after \ldots if one is listing a sequence. The authors don't do this in any of their lists, and this is very strange.

3) In the description of Algorithm 1, I'd change "S_max is denoted as" for "S_max denotes"

4) In (7) I think there is a "{" and a"}" missing before and after the o_{l,m_i}. It's set notation.

5) In the discussion, they write the word "differently." Would it be better to write "by contrast"?

**Experience Assessment:**

I have read many papers in this area.

**Review Assessment: Checking Correctness Of Derivations And Theory:**

I assessed the sensibility of the derivations and theory.

**Review Assessment: Checking Correctness Of Experiments:**

I assessed the sensibility of the experiments.

**Review Assessment: Thoroughness In Paper Reading:**

I read the paper at least twice and used my best judgement in assessing the paper.

---

> ### Author Response · Authors · 2019-11-11
> **Response to Reviewer #3**
>
> Thanks for your helpful comments.
> 1)"Matthew Effect" means that the rich get richer and the poor get poorer (wiki), originally. We use this description as an analogy to describe the unfairly training strategy in some weight sharing methods. In those methods, paths with better performance at early searching phases can obtain more training opportunities than other candidates, and further enhance their performance.
> 2) alpha is calculated according to Eq (4). We first normalize alpha to its softmax form p, after that a path will be dropped once its corresponding p is lower than th_\alpha. In our experiments, th_\alpha is set to a constant 0.5. We found that overall performance is largely robust (insensitive) to that parameter (th_\alpha), with less than 0.3% top1 variation when varying 0.4≤ th_\alpha ≤ 0.6. On one hand, when th_\alpha is set too high, some promising paths might be dropped at early stages, leading to degraded performance. On the other hand, when th_\alpha is set too low, paths will be dropped slow and we need more searching steps to keep the amount of finally remaining paths small enough. th_alpha is set to 0.5 to ensure that when trained to convergence, there are 2 or 3 paths remaining in each choice block.
> 3) Thanks very much for your constructive and detailed comments. We will fix the typos.

---

### Decision · Program_Chairs · 2019-12-19

**Decision:**

Reject

**Comment:**

This paper proposes a neural architecture search method that uses balanced sampling of architectures from the one-shot model and drops operators whose importance drops below a certain weight.

The reviewers agreed that the paper's approach is intuitive, but main points of criticism were:
- Lack of good baselines
- Potentially unfair comparison, not using the same training pipeline
- Lack of available code and thus of reproducibility. (The authors promised code in response, which is much appreciated. If the open-sourcing process has completed in time for the next version of the paper, I encourage the authors to include an anonymized version of the code in the submission to avoid this criticism.)

The reviewers appreciated the authors' rebuttal, but it did not suffice for them to change their ratings.
I agree with the reviewers that this work may be a solid contribution, but that additional evaluation is needed to demonstrate this. I therefore recommend rejection and encourage resubmission to a different venue after addressing the issues pointed out by the reviewers.